# Weight Space Learning for Certifiable Few-Shot Transfer Learning

## Abstract

In contemporary deep learning, a prevalent and effective workflow for solving low-data problems is adapting powerful pre-trained foundation models (FMs) to new tasks via parameter-efficient fine-tuning (PEFT). However, while empirically effective, the resulting solutions lack generalisation guarantees to certify their accuracy - which may be required for ethical or legal reasons prior to deployment in high-importance applications. In this paper we develop a novel transfer learning approach that is designed to facilitate non-vacuous learning theoretic generalisation guarantees for downstream tasks, even in the low-shot regime. Specifically, we first use upstream tasks to train a *distribution over PEFT parameters*. We then learn the downstream task by a *sample-and-evaluate* procedure – sampling plausible PEFTs from the trained diffusion model and selecting the one with the highest likelihood on the downstream data. Crucially, this confines our model hypothesis to a *finite* set of PEFT samples. In contrast to the typical continuous hypothesis spaces of neural network weights, this facilitates tighter risk certificates. We instantiate our bound and show non-trivial generalization guarantees compared to existing learning approaches which lead to vacuous bounds in the low-shot regime.

## 1 Introduction

Generalisation certificates are crucial for high-importance applications where accuracy should be guaranteed for legal or ethical reasons. Guarantees should certify the minimum testing accuracy expected on unseen data drawn from the training distribution. However, it is hard to establish non-trivial guarantees for large neural networks, since large learning capacity tends to produce looser guarantees. As such, there have only been a few successful demonstrations of non-vacuous guarantees for contemporary neural networks, even in the large-data regime (Dziugaite and Roy, 2017; Perez-Ortiz et al., 2021; Lotfi et al., 2024).

What about learning with sparse rather than large data? The problem of low-data learning is highly topical, due to the plethora of important limited-data applications (Wang et al., 2020), but challenging due to the difficulty of learning a large number neural network parameters without overfitting. This need has inspired several lines of research that make use of different forms of knowledge transfer, including meta-learning (Hospedales et al., 2021) and parameter-efficient transfer learning (PEFT) (Hu et al., 2021) from foundation models. While PEFT methods have recently been more empirically effective, neither family of approach has produced methods that can provide low-shot generalisation guarantees, to our knowledge. From a learning theoretic perspective this is because existing algorithms still search a hypothesis space (e.g., all neural network weights $\theta \in \mathcal{R}^N$) large enough to make known bounds vacuous when instantiated.

This paper introduces a novel approach to knowledge transfer that ultimately learns downstream tasks by *picking from a finite set of hypothesis*, where the set of hypothesis is fit to the upstream tasks. Our method, STEEL (Sample ThEn Evaluate Learner), facilitates using classic finite-hypothesis bounds, which are simple and tight, but not typically used in contemporary machine learning – which focuses on learning continuous value neural network parameters.

More specifically, in the upstream phase, we fit PEFT modules to available source tasks, and then train a parameter diffusion model to generate PEFTs according to this task distribution. In the downstream phase, we learn by sample-then-evaluate instead of traditional gradient descent. PEFT modules, unconditionally generated by the diffusion model, are scored using the target task training

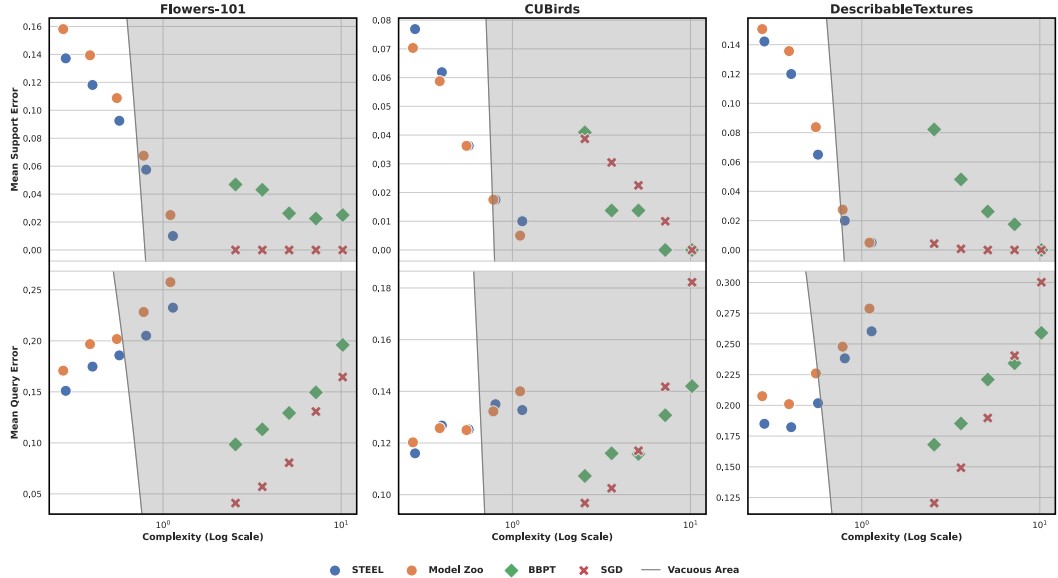

Figure 1: Generalization bounds for adapting CLIP to novel tasks (5-way classification with 1–16 examples per class). Plots show classification error (y-axis) versus the complexity term (x-axis, log scale; square root terms from Equations 3, and 7). Top/Bottom: Mean support/query (train/test) error on new tasks. Shaded regions indicate vacuous bounds, where (support error + complexity) $\geq 1$. Non-vacuous guarantees lie in the unshaded region. Competing methods (SGD, BBPT) fail to achieve non-vacuous bounds. In contrast, our method yields non-vacuous guarantees without significantly compromising training fit (top) or test accuracy (bottom).

set, and learning is to choose the highest scoring module. Compared to the original set of upstream models, the diffusion model can be more compact, and can interpolate between the original model set to achieve higher accuracy. This procedure is gradient-free, which has some scalability benefits (Malladi et al., 2023; Rezk et al., 2024), but more importantly it facilitates the use of PAC-Bayes finite-hypothesis bounds to provide non-vacuous guarantees, all while maintaining similar empirical accuracy to mainstream few-shot learning approaches. Figure 1 shows some illustrative results, demonstrating our learner's ability to maintain practical efficacy while being constrained to low enough complexity to provide non-vacuous guarantees (white zone).

In summary, our contributions are: (1) Introducing a novel learning paradigm for gradient-free transfer learning designed to facilitate accuracy guarantees for downstream tasks, even in the low-shot regime. (2) The first practical demonstration of non-vacuous generalization bounds for low-shot learning in large language and vision architectures.

## 2 RISK CERTIFICATES FOR DEEP MODELS

Certifying model generalization performance is fundamental in theoretical machine learning (Vapnik, 1995; Shalev-Shwartz and Ben-David, 2014; Mohri et al., 2018). Vapnik–Chervonenkis (VC), Rademacher, and PAC-Bayes bounds connect empirical risk (computable) to generalization risk (impossible to compute) through inequalities. Here we discuss the core concepts of risk certificates and why they are challenging to apply to deep models.

Let $h \in \mathcal{H}$ be a hypothesis (prediction function $y = h(x)$) and $\mathcal{H}$ the hypothesis space. In deep learning, $h$ corresponds to a model with parameters $\theta$, and $\Theta$ ($\ni \theta$) represents all possible parameter values, serving as the hypothesis space. A learning algorithm (i.e, SGD) selects $\theta$ from $\Theta$ given empirical data $S = (x_i, y_i)_{i=1}^{n}$ sampled i.i.d. from distribution $T$. The goal is minimizing generalization risk $R(\theta) = \mathbb{E}_{(x,y) \sim T}[l(\theta; x, y)]$, where $l(\theta; x, y)$ is the instance risk. Since computing $R(\theta)$ is impossible without access to $T$, we minimize empirical risk $r(\theta) = \frac{1}{n} \sum_{(x,y) \in S} l(\theta; x, y)$. Nevertheless, $r(\theta)$ is a surrogate for $R(\theta)$ and does not give any certificate about true generalization

risk. Therefore, theoretical bounds relate $R$ and $r$ as follows. For any $\theta \in \Theta$:

$$R(\theta) \leq r(\theta) + \text{ComplexityTerm}(\dim(\Theta), n) \tag{1}$$

where the complexity term depends on data size $n$ and hypothesis space complexity $\dim(\Theta)$, decreasing as $n$ increases and $\dim(\Theta)$ decreases[1]. The right side of (1) is the risk certificate, which guarantees an upper bound on the risk of generalization. All terms on the right side are also computable. Different bounding methods, for example VC and Pac-Bayes bounds, produce different complexity terms. In totality, certificates above 1 (with $l \in [0, 1]$) are vacuous, while we refer to those below 1 (or sometimes less than the random guess risk in the classification setting) as non-vacuous.

Traditional models like linear SVM achieve non-vacuous bounds through proper regularization (Vapnik, 1995; Shalev-Shwartz and Ben-David, 2014). However, conventional deep neural networks trained by gradient descent lack non-vacuous bounds unless $n$ is extremely large, due to high $\dim(\Theta)$ from numerous *continuous* parameters. Even sparse adapter methods (e.g., LoRA (Hu et al., 2021), LoRA-XS (Bałazy et al., 2024)) face challenges from the continuous nature of $\Theta$. No existing deep learning approach achieves meaningful generalization bounds in low-shot settings.

## 3 PROPOSED APPROACH

### 3.1 OUR APPROACH AT HIGH LEVEL

We propose a novel approach using a *finite* hypothesis space and gradient-free learning, departing from traditional continuous methods. In multi-task transfer learning, we learn a task distribution via a diffusion generative model, creating a *finite* hypothesis space $\Theta$ from model-generated samples. Our simple learning algorithm selects $\theta \in \Theta$ with minimal empirical risk, using heuristic search for efficiency with large $\Theta$ to reduce the forward-pass overhead (as detailed in Sec. 3). The use of a diffusion model in STEEL is motivated by recent work on generative modeling in weight spaces (Wang et al., 2024), which demonstrates that diffusion models can effectively capture the complex, multimodal distribution of neural network parameters.

This approach, combined with finite-hypothesis PAC-Bayes bounds, yields tight non-vacuous risk certificates on large-scale LLM/vision benchmarks using FLAN-T5/CLIP models. Importantly, test performance remains comparable to standard learning algorithms. The *evaluate-then-select* strategy optimizes PAC-Bayes bounds by keeping the complexity term constant. We present the formal problem setup below before detailing our approach in Sec. 3.3.

### 3.2 PROBLEM SETUP AND NOTATION

We consider *low-shot cross-task transfer learning*, where we are given training tasks $T_1, \ldots, T_N$, i.i.d. from an unknown task distribution $p_{\text{true}}(T)$. At test time, we receive a new task $T^* \sim p_{\text{true}}(T)$, observed only via a small support set $S^* = \{(x_i, y_i)\}_{i=1}^n$. This setting aligns with assumptions made in meta-learning (Hospedales et al., 2021) and model-zoo approaches (Huang et al., 2024).

Our goal is to find a way to ensure tight generalization bounds for the underlying deep models at test time. As discussed in Sec. 2, a main challenge here is that we have low-shot data $S^*$ and a large number of model parameters, where the latter immediately translates into high hypothesis space complexity. Hence, applying the traditional learning theories directly to this problem leads to vacuous error bounds. We come up with a new method that exploits the training tasks to transfer the knowledge to unseen tasks so that it can offer tight non-vacuous risk certificates.

### 3.3 TRANSFER LEARNING BY SAMPLE-AND-EVALUATE

Our first observation is that gradient-based model adaptation to low-shot samples $S^*$ must be avoided to reduce hypothesis space complexity (Sec. 2). Our key intuition is that we can learn the task distribution $p_{true}(T)$ from training tasks $\{T_i\}_{i=1}^N$, but doing so introduces a strong inductive bias

---

[1]PAC-Bayes bounds compute expected risk over a posterior on $\theta$ and include a complexity term from divergence to a prior. If we confine the posterior to concentrate at a point and use a flat prior, the form resembles (Eq.1). Some bounds relate $R$ and $r$ nonlinearly, but can be approximated by (Eq.1) without affecting our argument.

or regularization. Let $\theta_i$ be the learned neural network parameters for task $T_i$. (Throughout, we treat PEFT adapter parameters as $\theta$, keeping the pre-trained backbone fixed.) We view $\theta_i$ as the best description of $T_i$ and collect task-wise parameters $\{\theta_i\}_{i=1}^N$. Learning $p_{true}(T)$ thus reduces to a *density estimation problem*, i.e., estimating $p(\theta)$ from i.i.d. samples $\{\theta_i\}_{i=1}^N$, treating $p(\theta)$ as a surrogate for $p_{true}(T)$.

We learn a diffusion model $p(\theta)$ with $\{\theta_i\}_{i=1}^N$ as training data following Ho et al. (2020). At test time, the estimated $p(\theta)$ serves as a proxy for $p_{true}(T)$. We generate plausible candidate samples $\theta$ (i.e., tasks $T$) and select the one closest to $T^*$. Since only $S^*$ of $T^*$ is available, we choose the sample with the least discrepancy from $S^*$ via the minimum loss rule: $\arg\min_{\theta \in \Theta} \sum_{(x,y) \in S^*} l(\theta; x, y)$, where $\Theta$ is the set of diffusion model candidates. This corresponds to empirical risk minimization where $\Theta$ acts as the *hypothesis space*, and we call this learner STEEL (Sample ThEn Evaluate Learner).

This strategy amounts to *selection from a finite hypothesis space*, rather than searching a continuous space via gradient-based fine-tuning. The choice of a finite hypothesis set (diffusion model samples) for learning enables a strong regularizing inductive bias to be learned from upstream tasks.

### 3.4 RISK CERTIFICATE WITH FINITE HYPOTHESIS SPACE

Suppose we have a well-trained set of PEFT adapter parameters $\overline{\Theta} = \{\theta_i\}_{i=1}^N$, where each $\theta_i$ is optimal for the $i$-th training task. Without diffusion, a natural baseline is to use the model zoo $\overline{\Theta}$ directly as the hypothesis space, i.e., $\Theta = \overline{\Theta}$. This becomes increasingly reasonable as $N$ grows, since $\overline{\Theta}$ more closely approximates the true task distribution $p_{true}(T)$. We refer to this as the *model zoo* strategy. Our full STEEL method goes further: we train a diffusion model $p(\theta)$ on $\overline{\Theta}$ and define $\Theta$ as samples drawn from this model.[2] While both are our proposals, the diffusion-based approach is our primary strategy, as it offers two key advantages: (i) scalability—storing a single diffusion model is far more efficient than keeping $N$ separate adapters; and (ii) generalization—diffusion models are known to interpolate well between training samples, enabling better approximation of $p_{\text{true}}(T)$ and often leading to improved test accuracy.

Few-shot adaptation is done by evaluate-then-select:

$$\theta^* = \arg\min_{\theta \in \Theta} r(\theta) = \frac{1}{n} \sum_{(x,y) \in S^*} l(\theta; x, y) \qquad (2)$$

We expect that $\Theta$ is rich enough to represent the true task distribution $p_{true}(T)$ faithfully, and the adapted ("selected") $\theta^*$ will generalize well on unseen samples from $T^*$.

A crucial benefit of our test-time adaptation strategy (2) is that we have a tight provable generalization error bound that can serve as a risk certificate for its test-time prediction quality. This mainly originates from the *finite* hypothesis space $\Theta$. More specifically, using the PAC-Bayes theorems (e.g., Sec. 2.1.3 in (Alquier, 2021)), we can show that with probability at least $1 - \epsilon$,

$$R(\theta) \leq r(\theta) + C \cdot \sqrt{\frac{\log \frac{|\Theta|}{\epsilon}}{2n}} \quad \text{for any } \theta \in \Theta \qquad (3)$$

where $R(\theta) = \mathbb{E}_{(x,y) \sim T^*}[l(\theta; x, y)]$ is the generalization error of $\theta$, and $C$ is the maximal loss value (i.e., $0 \leq l \leq C$). The bound immediately comes from the PAC-Bayes theorem with the (data-independent) uniform prior over $\Theta$ and the Dirac's delta posterior choice. Since the size of the hypothesis space $|\Theta|$ only appears in the $\log$ term, a massively large $\Theta$ is allowable while retaining a tight bound. Furthermore, the bound can be minimized with the smallest empirical error $r(\theta)$, i.e., $\theta = \theta^*$, which justifies our evaluate-then-select strategy (2).

However, the computational question naturally arises: *How do we solve (2) efficiently?* We consider two solutions:

- **Exhaustive search.** Evaluate $r(\theta)$ for all $\theta \in \Theta$ and select the one with the lowest loss. This guarantees finding the optimal $\theta^*$, but is often computationally intractable (e.g., LLMs require a prohibitive number of forward passes or generations due to large $|\Theta|$).

---

[2]Alternatively, one could augment $\Theta$ with both zoo and diffusion samples. We focus on comparing the pure zoo and diffusion strategies to highlight their differences.

- **Hierarchical search.** Use a tree-based approach such as hierarchical clustering of $\Theta$. Evaluate losses at the top level (e.g., cluster centroids or medoids), select the best cluster, and recurse within it. This reduces computation to $O(\log |\Theta|)$ and can yield a good approximation to $\theta^*$, but may result in suboptimal empirical loss $r(\theta)$ due to early pruning.

We reiterate that the generalization bound (3) holds across all strategies, though suboptimal $\theta$ choices may slightly increase empirical loss $r(\theta)$ and loosen the certificate. We provide STEEL pseudo-code in Appendix D.

## 4 RELATED WORK

**Risk certificates.** While traditionally applied to simple models, recent work extends risk certificates to deep learning. Notable approaches include using data-dependent priors (Perez-Ortiz et al., 2021) and parameter quantization of PEFT adapters (Lotfi et al., 2024) in tabula-rasa and single source transfer learning respectively. Nevertheless, these still require large training sets to obtain non-vacuous bounds. In the meta-learning literature, Zakerinia et al. (2024) proposed meta-learning PAC-Bayes bounds, which aims to facilitate tighter generalisation bounds by extracting a common prior from up-stream tasks in a multi-task setting. However, it was only demonstrated on toy problems and doesn't scale to large models due to the memory requirements of nested gradient computations.

**Model diffusion methods.** Several works explore diffusion for generating model parameters: NNDiffusion (Wang et al., 2024) for BN modules, ProtoDiff (Du et al., 2023) for ProtoNet-based few-shot learning, and MetaDiff (Zhang et al., 2024) and D2NWG (Soro et al., 2025) for gradient-free meta-learning. Scaling properties of Diffusion based learning were also explored (Schürholt et al., 2024). However, none provide risk certificates for the generated models.

**Sparse adapter (PEFT) methods.** Sparse adapters reduce learnable parameters, crucial for our diffusion-based sampling. While LoRA (Hu et al., 2021) uses trainable low-rank matrices and VeRA (Kopiczko et al., 2024) uses fixed matrices with trainable diagonals, we adopt LoRA-XS (Bałazy et al., 2024), which uses SVD with a trainable full matrix for the singular values. Such adapters have facilitated large-data guarantees (Lotfi et al., 2024), but in our framework they will facilitate low-shot guarantees.

**Connection to our Approach.** In terms of assumptions, we consider the same multi-task setting of meta-learning (Hospedales et al., 2021), model-zoo (Huang et al., 2024) and model diffusion (Wang et al., 2024) approaches. They all aim to facilitate downstream learning by knowledge transfer from upstream tasks. In terms of solution we share the benefit of model-zoo and model-diffusion methods in being able to use third-party off-the-shelf pre-trained upstream models, rather than requiring an expensive joint learning procedure like meta-learners. Like meta-learners and model-diffusers, but unlike zoo methods, we do learn a task-agnostic component (the diffusion model).

## 5 EXPERIMENTS

**Metrics for Certified Learning Quality:** In the low-shot learning context, we run many learning episodes with different random small training sets (Wang et al., 2020). Thus we need evaluation metrics for the typical empirical performance and certificate strength instead of a single accuracy and certificate for one large scale learning (Lotfi et al., 2024). This is straightforward for empirical test performance: we report the average over episodes of the relevant task metric (e.g., accuracy, RMSE). For certificates, we report the minimum (best), median (typical) and maximum (worst) case error guarantee over all downstream tasks. Since many learners often produce vacuous certificates in the low-shot regime (e.g., guaranteed error not below 1), we use *the proportion of tasks which have a non-vacuous certificate* as our leading metric. Finally, to quantify how tight the certificates are, we also report *gap* - the average distance between certificates and query errors over all faithful bounds. More concretely, this is computed as $\sum_{i=0}^{K} \frac{\max(0, \text{bound}_i - \text{query error}_i)}{K}$ where $K$ is the total number of downstream tasks.

**Competitors:** For single task methods that exploit only the foundation model and the target downstream task, we compare standard SGD, along with MeZO (Malladi et al., 2023) and BBPT (Yu et al., 2023) – two state of the art gradient-free learners – for fine-tuning. For multi-task alternatives that

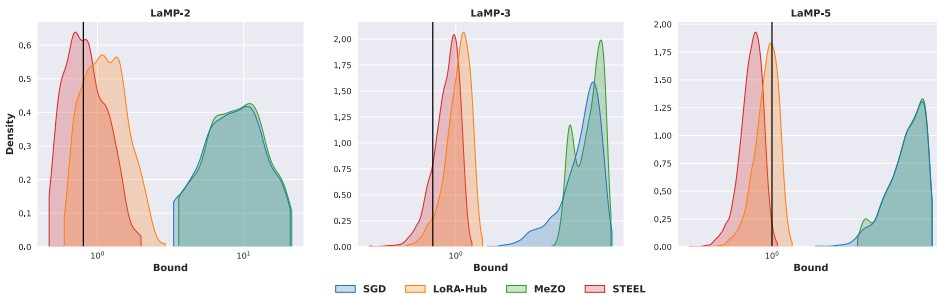

Figure 2: Distribution of generalisation guarantees (x-axis, log scale) obtained over few-shot LLM adaptation episodes. Vertical lines indicate the vacuous bound threshold. STEEL provides a dramatically better distribution of provable generalisation outcomes compared to alternatives.

exploit also the same task distribution as STEEL, we compare the model zoo learner LoRA-Hub (Huang et al., 2024) and the meta-learner MetaPB (Zakerinia et al., 2024). LoRA-Hub randomly samples from the model zoo and learns a new adapter as a linear combination of sampled adapters. Note that all methods are already combined with the same (experiment-specific) PEFT strategy for fair comparison. For SGD, MeZO, BBPT and MetaPB, this is an upgrade provided by us - they would definitely otherwise fail in the full parameter space. For SGD, MeZO, BBPT and LoRA-Hub, we combine them with recent quantization bound of Lotfi et al. (2024) (Eq. 7) to provide guarantees, MetaPB uses their own PAC Bayes bound, and STEEL uses the bound in Eq. 3.

## 5.1 GUARANTEES FOR FEW-SHOT LLM ADAPTATION

**Datasets:** We use the LaMP personalization benchmark (Salemi et al., 2024) for low-shot LLM adaptation. LaMP consists of fixed training (seen) and evaluation (unseen) clients, each with support and query data. We evaluate on three datasets: LaMP-2 (nominal classification: movie tagging), LaMP-3 (ordinal classification: product rating), and LaMP-5 (text generation: scholarly title generation).

**Setup:** Following prior work (Tan et al., 2024; Salemi et al., 2024), we fine-tune Flan-T5 base (Chung et al., 2024) on all seen users' support data to build a task-specific base model. Then, we personalize it by training a LoRA-XS module per user (Bałazy et al., 2024), with rank 6 and alpha 16 (2592 tunable parameters). These modules, trained on the same support data used for base fine-tuning, form our model zoo.

Table 1: LaMP LLM adaptation benchmark results.

| | | SGD | LoRA-Hub | MeZO | STEEL |
|---|---|---|---|---|---|
| **LaMP-2** | % Non-Vacuous Tasks | 0.00% | 32.13% | 0.00% | **65.12%** |
| | Median Gap | 8.12 | 0.68 | 8.45 | **0.43** |
| | Min Bound | 3.32 | 0.59 | 3.60 | **0.47** |
| | Median Bound | 8.52 | 1.12 | 8.85 | **0.80** |
| | Max Bound | 20.86 | 2.90 | 21.36 | **1.99** |
| | Accuracy↑ | 63.25% | 57.51% | 63.30% | 63.74% |
| | F1↑ | 56.15% | 50.84% | 57.03% | 55.69% |
| **LaMP-3** | % Non-Vacuous Tasks | 0.00 | 5.00 | 0.00 | **15.48** |
| | Median Gap | 3.09 | 0.23 | 3.10 | **0.14** |
| | Min Bound | 1.35 | 0.51 | 2.54 | **0.43** |
| | Median Bound | 3.56 | 1.04 | 3.76 | **0.93** |
| | Max Bound | 4.75 | 1.30 | 4.53 | **1.17** |
| | MAE↓ | 0.217 | **0.230** | 0.242 | 0.231 |
| | RMSE↓ | 0.511 | 0.526 | 0.531 | 0.524 |
| | Cross-Entropy↓ | 0.479 | 0.739 | **0.626** | 0.693 |
| **LaMP-5** | % Non-Vacuous Tasks | 0.00 | 63.84 | 0.00 | **99.16** |
| | Median Gap | 4.04 | 0.41 | 4.04 | **0.26** |
| | Min Bound | 1.61 | 0.52 | 2.58 | **0.40** |
| | Median Bound | 4.57 | 0.96 | 4.57 | **0.81** |
| | Max Bound | 5.86 | 1.25 | 5.88 | **1.06** |
| | ROUGE-1↑ | 47.04% | 47.05% | 47.03% | **47.22%** |
| | ROUGE-L↑ | 42.79% | 42.75% | 42.73% | **42.89%** |

For efficiency, STEEL uses hierarchical search (Section 3.4). Hyperparameters are in Appendix B.2.

**Results:** Our main contribution relates to the ability to provably certify the generalisation of low-shot learning. In terms of low-shot LLM adaptation, Figure 2 visualises the distribution of certification outcomes over a large number of episodes for the three LAMP benchmarks. Taking note of the log-scale on the x-axis for generalisation guarantee strength, we can see that our STEEL learner provides dramatically better guarantees than conventional continuous-parameter learner alternatives, thanks to its discrete hypothesis space. The vertical lines indicate the threshold for vacuous bounds. Standard learners such as SGD and MeZO have no mass left of the threshold, while a substantial number of STEEL learning episodes are non-vacuously guaranteed. Table 1 provides more detailed quantitative results in terms of various metrics. Notably, to assess provable generalisation, the data visualised in Figure 2 is summarised as the % non-vacuous metric (the fraction of episodes which have guarantees with a strength above chance-level), and the median guarantee strength across episodes.

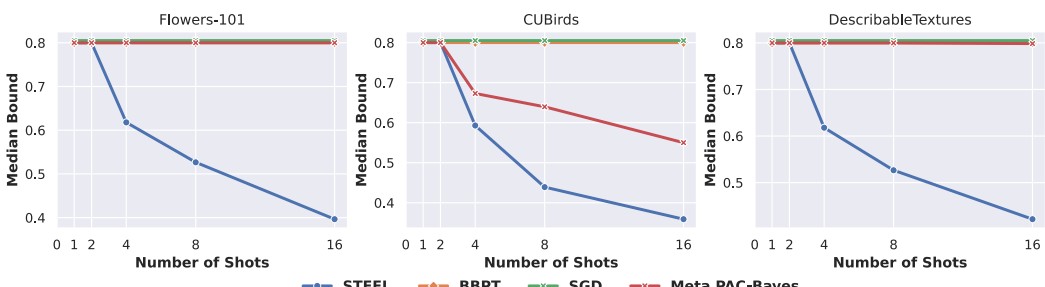

Figure 3: Dependence of generalisation guarantee on training set size. Our finite-hypothesis class learner STEEL achieves non-vacuous guarantees from 4-shot onward. Standard approaches provide no guarantees anywhere in this low-shot range.

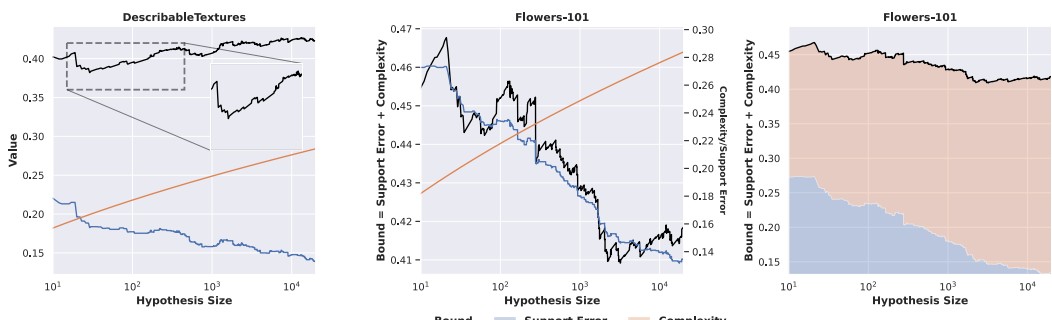

Figure 4: "Learning curves" illustrating empirical and certified learning dynamics of STEEL with respect to samples/iterations, which is equivalent to hypothesis space size. More samples improves the training (support) error, while increasing the complexity penalty. The sum of these two terms instantiates the generalisation guarantee (Eq. 3) achieved for a given number of samples.

From Table 1 results we can see that: (1) Standard supervised learning approaches such as (gradient-based) SGD and (gradient-free) MeZO have no non-vacuous episodes - no few-shot learning task can be guaranteed. (2) STEEL has the most non-vacuous episodes for each benchmark, with almost every few-shot learning episode being guaranteed in the LAMP-5 benchmarks. The median STEEL episode also has a substantially non-vacuous guarantee for all three benchmarks. (3) Interestingly, LoraHUB combined with Lotfi et al. (2024)'s discretization bound also has some non-vacuous episodes, but less than STEEL. (3) STEEL has comparable or better empirical test accuracy compared to existing approaches such as SGD and MeZO, while providing a huge improvement in certifiability.

## 5.2 GUARANTEES FOR FEW-SHOT VISUAL RECOGNITION

**Datasets:** We use fine-grained vision datasets with standard seen/unseen class splits: CUBirds (Wah et al., 2011), FGVC-Aircraft (Maji et al., 2013), Describable Textures (Cimpoi et al., 2014), and Flowers-101 (Nilsback and Zisserman, 2008). Splits follow Meta-Dataset (Triantafillou et al., 2020) for Flowers and Learn2Learn (Arnold et al., 2020) for the others.

**Setup:** We sample $k$-shot, 5-way tasks following meta-learning conventions (Triantafillou et al., 2020), evaluating at $k \in \{1, 2, 4, 8, 16\}$. CLIP (Radford et al., 2021) serves as the backbone. For each task, we sample 5 classes, draw $n$ support examples per class, and adapt using CoOp (Zhou et al., 2022)—i.e., prompt tuning with a 2-token prefix per class (1024 total tunable parameters). The model zoo is built from training classes; evaluation uses disjoint unseen classes. Due to fast inference, we perform exhaustive search over sampled prompts. Meta-training, evaluation, and bound details for Meta-PB are provided in Appendix A.3. Hyperparameters for all other vision experiments are in Appendix B.3.

**Results:** The results in terms of mean training and testing error versus complexity are summarised for three datasets in Figure 1. The dots for each learner reflect the training set sizes of 1, 2, 4, 8,

and 16-examples per-class, and the white/grey zone separation delineates the space of non-vacuous vs vacuous bound outcomes. The main message is that only our finite hypothesis class approaches achieve any non-vacuous guarantees across this whole range of training set sizes. Every result for the standard SGD and BBPT approaches is vaccuous and cannot be guaranteed. Please note, Meta-PB is not included as it is not directly comparable on the plot y-axis (empirical error term); Meta-PB uses the unbounded cross-entropy loss (Equation 20) as the risk metric, while our and other competing methods use empirical error.

For the 16-shot case, these experiments are quantified in Table 2. Similarly to the results for LLM adaptation, we can see the dramatic difference in % of non-vacuous episodes, and dramatic improvement in the min, median and max bound obtained over episodes. Compared to the LLM case, STEEL pays a slightly higher price in terms of empirical test accuracy compared to SGD for some benchmarks, however this is small compared to the stark difference in certification performance. The state of the art Meta-PB, also oriented at low-shot certification in some cases (CUB) manages to non-vacuously certify most episodes. However, in all cases the strength of this certification is much worse than STEEL (gap, median bound).

Figure 3 highlights the evolution of the median generalisation bound as a function of the training set size. For STEEL it becomes non-vacuous from 4-shots onward, and the standard approaches never become non-vacuous[3]

### 5.3 FURTHER ANALYSIS

**Learning Curves:** We discuss and provide some insight into the learning process of our discrete hypothesis class learner. Standard gradient-descent takes repeated update steps to find a model that better fits a training set. By analogy, our gradient-free STEEL draws more samples as it attempts to iteratively sample a model that better fits the training set. Our main experiments use a fixed number of 20,000 samples on all vision datasets. Figure 4 illustrates our learner's behaviour by showing the equivalent of a learning curve for our model. The x-axis is the number of samples drawn, and equivalently the learning theoretic hypothesis space size. Unlike SGD, this means that there is a direct dependence of hypothesis class complexity ($|\Theta|$ in Eq. 3) and the number of iterations/samples. This is reflected in the steadily increasing red complexity curve in Figure 4(left, middle). We

Table 2: Aggregate over 16-Shots 5-way visual recognition episodes. Standard Errors are reported in Appendix C.3

| Method | SGD | BBPT | Meta-PB | STEEL |
|---|---|---|---|---|
| **CUBirds** | | | | |
| Non-Vacuous Ratio | 0.00% | 0.00% | 97.50% | **100.00%** |
| Average Gap | 2.49 | 2.48 | 0.45 | **0.24** |
| Min Bound | 2.55 | 2.55 | 0.34 | **0.30** |
| Median Bound | 2.58 | 2.59 | 0.55 | **0.36** |
| Max Bound | 2.64 | 2.68 | 0.85 | **0.47** |
| Average Accuracy | 90.32% | **89.27%** | 89.24% | 88.40% |
| **Describable Textures** | | | | |
| Non-Vacuous Ratio | 0.00% | 0.00% | 50.00% | **100.00%** |
| Average Gap | 2.43 | 2.46 | 0.54 | **0.24** |
| Min Bound | 2.55 | 2.55 | 0.57 | **0.36** |
| Median Bound | 2.55 | 2.63 | 0.80 | **0.42** |
| Max Bound | 2.58 | 2.78 | 1.06 | **0.53** |
| Average Accuracy | 87.95% | **83.20%** | 81.12% | 81.50% |
| **FGVCAircrafts** | | | | |
| Non-Vacuous Ratio | 0.00% | 0.00% | 0.00% | **97.50%** |
| Average Gap | 2.31 | 2.45 | 0.86 | **0.22** |
| Min Bound | 2.58 | 2.64 | 0.87 | **0.45** |
| Median Bound | 2.64 | 2.80 | 1.25 | **0.61** |
| Max Bound | 2.78 | 3.01 | 1.76 | **0.85** |
| Average Accuracy | 65.57% | **62.37%** | 59.84% | 61.37% |
| **Flowers-101** | | | | |
| Non-Vacuous Ratio | 0.00% | 0.00% | 10.00% | **100.00%** |
| Average Gap | 2.51 | 2.50 | 0.68 | **0.27** |
| Min Bound | 2.55 | 2.55 | 0.65 | **0.31** |
| Median Bound | 2.55 | 2.59 | 1.15 | **0.40** |
| Max Bound | 2.55 | 2.70 | 1.77 | **0.61** |
| Average Accuracy | 95.90% | **90.15%** | 71.23% | 84.90% |

can also see that the training/support error goes down consistently over iterations/samples as the sampler progressively discovers better models. The generalisation bound (black line) is given by the sum of the training error and complexity. The figure illustrates one case (Flowers, middle, right) where the bound continues to improve up to a large number of samples/hypothesis size, because the continued improvement in training error outweighs the complexity gain. It also illustrates a case (DTD, left) where the training error improvement is slower and quite rapidly outweighed by the complexity gain, so that the best bound is actually achieved after quite a small number of samples.

**Sampler vs Zoo:** STEEL compresses the upstream pre-trained models into a learned model generator. Selecting from the upstream models using downstream task performance as a criterion provides an alternative approach to learning that also corresponds to a finite hypothesis space. Our generator approach was motivated by ensuring scalability with respect to a large number of upstream models, and also to improve accuracy by enabling interpolation between upstream models rather than solely

---

[3]Note their bound is substantially worse than 0.8, but for simple visualisation, we plot it as chance-level for 5-way classification.

being limited to selecting one of them. Figure 1 shows that STEEL's diffusion sampler tends to provide improved accuracy compared to its raw model zoo, especially for Flowers and DTD. On average across all datasets STEEL consistently outperforms Model Zoo. Detailed per-dataset per-shot performance is deferred to Appendix C, Table 4. Furthermore, we show in Appendix C.1 how to certify zero-shot CLIP using confidence intervals, and demonstrate that STEEL not only achieves tighter guarantees but also substantially outperforms it in terms of empirical performance.

**Tightening STEEL Certificates:** So far, we have focused on the very low-shot regime to highlight our key result: the first non-vacuous certificate in this challenging setting. However, many applications may require stronger guarantees. To explore this, we increase the support set size from very low to moderately low shot using the iNaturalist dataset (Horn et al., 2018) (birds subset, excluding overlapping classes), with a diffusion model pre-trained on CUB. This setup allows scaling up to 128-shot evaluation. Figure 5 shows: (1) As the training set grows, the support error increases initially as there are more data points to fit, but soon stabilizes. The finite hypothesis-space does not problematically under-fit an increasingly large training set. (2) Query error decreases and saturates around 64

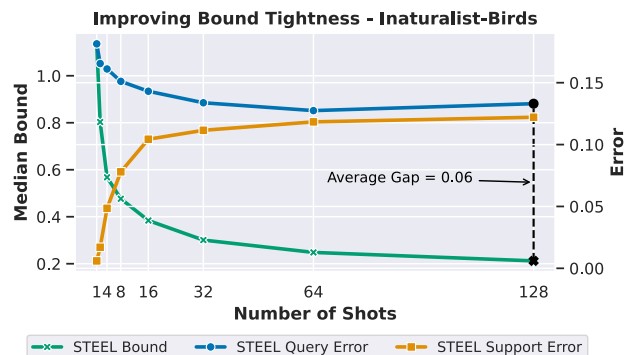

Figure 5: **Support/query error and certified risk vs. support set size on iNaturalist birds.** As the number of shots increases, support and query errors converge, while the bound continues to tighten. The gap between the certified risk and empirical query error drops to 6% at 128 shots, demonstrating the ability to produce tight certificates.

shots. (3) Crucially, the STEEL bound keeps tightening: as support size grows, the complexity term continues to shrink even after support error plateaus, resulting in a certified risk just 6% above empirical test error at 64–128 shots—while using far fewer examples than standard certification approaches (Lotfi et al., 2024). (4) Finally, it is important to note that this strong result is *despite* the upstream-downstream distribution shift (CUB → iNaturalist), showing that STEEL is robust to some task distribution shift.

**Diffusion Ablation:** Appendix E presents ablation experiments comparing our diffusion model to alternative generative approaches. Results show that diffusion outperforms these baselines, supporting its use as the preferred generative model in our method.

## 6 LIMITATIONS AND FUTURE WORK

As discussed in the main paper (see Section 5.3), increasing support set size continues to tighten the certificate without causing underfitting. Beyond this, there remain several promising directions for further improving certificate strength. One avenue is to refine the structure of the hypothesis class. The current bound treats hypotheses as independent, but in practice, many may yield highly correlated predictions. Accounting for this redundancy, for instance, by discounting the contribution of similar models, could lead to tighter bounds. Another opportunity lies in the use of hybrid hypothesis classes. For example, treating each sample as a mean of a Gaussian mixture component and applying PAC-Bayesian fine-tuning may yield posterior distributions that better fit the data while maintaining low complexity. These strategies suggest that the certification framework presented here can be further enhanced by incorporating richer modeling and more expressive prior structures.

Finally, recent advances in weight-space learning architectures provide additional opportunities to boost the empirical performance of STEEL. Architectures such as SANE Schürholt et al. (2024) are natural candidates for integration into the STEEL plug-and-play pipeline. This is possible because the applicability of the certificates does not depend on the specific diffusion model architecture. What matters instead is the empirical risk of the generated samples, which directly influences certificate tightness.

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

# A  APPENDIX

## A  EXISTING RISK BOUNDS FOR DEEP MODELS

### A.1  VANILLA PAC-BAYES BOUND

This is the vanilla, non-transfer learning bound. As a baseline, one can also contrast with the vanilla PAC-Bayes bound (i.e., non-meta learning bound). This essentially follows the Cantoni's bound, and can be written as follows. With probability at least $1 - \epsilon$, the following holds:

$$\mathbb{E}_{Q(\theta)}[R(\theta)] \ \leq \ \mathbb{E}_{Q(\theta)}[r(\theta)] + \sqrt{\frac{\mathrm{KL}(Q||\pi) + \log(1/\epsilon)}{2n}} \tag{4}$$

Here $n$ is the test support size. We can set $\pi = \mathcal{N}(0, \kappa^2 I)$ for some fixed $\kappa_\pi$ and $Q(\theta) = \mathcal{N}(\mu, \Sigma^2)$ where the parameters $(\mu, \Sigma)$ can be learned by minimizing the right hand side. The sampled version $\theta^z = \mu + \sqrt{\Sigma} \cdot z$, $z \sim \mathcal{N}(0, I)$ can be used during the optimization. Once optimized, the minimum value of the right hand side serves as the error bound for the test task.

### A.2  BOUND WITH PARAMETER-LEVEL QUANTIZATION

In (Lotfi et al., 2024), they proposed a non-vacuous bound for the LLM based on the model parameter quantization (e.g., fixed-length floating point machine representation). There are several differences to our approach:

1. The paper is about LLM pre-training setup with large training data, and the bound would be vacuous if training data size is not large enough (e.g., $\geq 10K$).

2. They derive the same finite hypothesis space PAC-Bayes bound, but replace the $\log |H|$ term by $\log(1/p(h))$ where $p(h)$ is the prior likelihood, and $\log(1/p(h))$ is approximated and upper-bounded by $C(h)$ which is the number of bits for representing the hypothesis $h$.

3. The finite hypothesis space comes from the fixed-size floating point representation for real numbers (e.g., if there are $d$ trainable parameters, then $C(h) = d \cdot 32$), but to reduce it further, they propose what is called the SubLoRA, which is a random subspace representation (i.e., $\theta = Pw$, $P$ = random subspace basis, $w$ = coefficients) of the LoRA $A/B$ matrices.

4. Also, instead of 32 bit for each of $d$ params, they do some clustering to reduce it to shorter coding, more precisely the arithmetic coding.

The followings are some details of their bound derivation. With probability at least $1 - \epsilon$,

$$R(\theta) \leq r(\theta) + C \cdot \sqrt{\frac{K(\theta) + 2\log K(\theta) + \log(1/\epsilon)}{2n}} \tag{5}$$

where $K(\theta)$ is the Kolmogorov complexity bound that can be estimated as:

$$K(\theta) = \sum_{i=1}^{d} (\# \text{ of bits in the arithmetic coding of } \theta_i) \tag{6}$$

where $d = \dim(\theta)$ for the PEFT parameters $\theta$. The arithmetic coding requires clustering of parameters $\theta_i$s, thus being dependent on the particular $\theta$ used. However, we can consider the best (i.e., the tightest) bound possible. That is, even if we have 1 bit for every $\theta_i$ (the minimal code length possible), $K(\theta) = d$, and plugging this into (5) yields:

$$R(\theta) \leq r(\theta) + C \cdot \sqrt{\frac{d + 2\log d + \log(1/\epsilon)}{2n}} \tag{7}$$

which is the *best* scenario. Please note that this should be interpreted in the context of currently known bounds rather than all possible bounds.

In (5), as before, $R(\theta) = \mathbb{E}_{z \sim T^*}[l(\theta; z)]$ is the generalization error of $\theta$, $r(\theta) = \frac{1}{n} \sum_{z \in S^*} l(\theta; z)$ is the empirical error on the support data with size $n = |S^*|$, and $C$ is the maximal loss value (i.e., $0 \leq l \leq C$).

### A.3 PAC-BAYES META LEARNING BOUND

In (Zakerinia et al., 2024) they proposed an extension of the PAC-Bayes bound for meta learning. Their meta learning algorithm aims to learn a distribution over the adaptation algorithms $A(S) \rightarrow Q(\theta)$ where $S$ is the support training data for a task and $Q(\theta)$ is the posterior distribution in the PAC-Bayes theorem. In their paper, any adaptation algorithm $A$ is allowable provided that $A$ internally makes use of a prior distributions in the PAC-Bayes theorem (e.g., the adaptation algorithms that try to minimize the PAC-Bayes bound), so that $A(S)$ becomes a function of the prior. And the meta learning model is a distribution $\rho$ that can sample the prior, and the meta learning amounts to learning the distribution $\rho$.

Although they provided a concrete example in Appendix B.2 in their paper, the described meta training involves an optimization of a loss function that contains an optimal value of another problem, which is typically attained by gradient descent. Just like MAML-type algorithms with a large number of inner gradient steps, this renders the application of their theorem to large-scale model/data impractical. So we propose a more practical reptile-like adaptation strategy, essentially admitting a closed-form inner optimal solution, that is practical in large model/data scenarios. Note that under our setting the meta learning PAC-Bayes bound is still applicable since they said the theorem works for *any* adaptation algorithms.

#### CONCRETE CASES WITH (REPTILE-LIKE) QUADRATIC-REGULARIZED ADAPTATION ALGORITHMS

Let $\theta$ be the parameters of a neural network or PEFT parameters. Denote $\dim(\theta)$ by $d$. The adaptation algorithm $A(S)$ returns a PAC-Bayes posterior $Q(\theta) = \mathcal{N}(\mu, \Sigma)$ for some task support data $S$ as input. We consider diagonal $\Sigma$. The algorithm $A$ itself is parametrized by the PAC-Bayes prior $P_1(\theta) = \mathcal{N}(\mu_1, \Sigma_1)$ since we only deal with those $A$s that internally use $P_1$ as a part of the algorithm without dependency on any other things. One candidate for $A$ is the algorithm that minimizes the PAC-Bayes bound with prior $P_1$ with respect to $Q(\theta)$. However, as said earlier, this incurs MAML-like optimization that is impractical.

Our choice of $A$ is as follows: (Step-1) We first find $(\mu^{SGD}, \Sigma^{SGD}) = \arg\min_{\mathcal{N}(\theta; \mu, \Sigma)} \mathbb{E}[l(\theta; S)]$ by gradient descent. Technically, this involves evaluating $l(\theta; S)$ with reparametrized $\theta = \mu + \sqrt{\Sigma}z$ for $z \sim \mathcal{N}(0, I)$, where the gradients take: $\nabla_\mu l(\theta; S) = \nabla_\theta l(\theta; S)$ and $\nabla_\Sigma l(\theta; S) = \nabla_\theta l(\theta; S) \cdot 0.5\Sigma^{-0.5}z$ by the chain rule. So it is as tractable as SGD since we only need conventional backprop $\nabla_\theta l(\theta; S)$ to get gradients with respect to $(\mu, \Sigma)$. (Step-2) Solve the following quadratic optimization:

$$(\mu^*, \Sigma^*) = \arg\min_{\mu, \Sigma} ||(\mu, \Sigma) - (\mu^{SGD}, \Sigma^{SGD})||^2$$
$$+ \alpha \cdot ||(\mu, \Sigma) - (\mu_1, \Sigma_1)||^2 \tag{8}$$

The intuition is that the adapted model $(\mu, \Sigma)$ has to be close to the fitted $(\mu^{SGD}, \Sigma^{SGD})$ and also close to the prior $(\mu_1, \Sigma_1)$. The trade-off coefficient $\alpha$ is user's choice, but when inspired by the PAC-Bayes bound objective, we can set $\alpha = 1/\sqrt{|S|}$. The main benefit of this quadratic-regularized adaptation algorithm is that we have the closed-form solution:

$$\mu^*(\mu_1, S) = \frac{1}{1+\alpha}\mu^{SGD} + \frac{\alpha}{1+\alpha}\mu_1 \tag{9}$$

$$\Sigma^*(\Sigma_1, S) = \frac{1}{1+\alpha}\Sigma^{SGD} + \frac{\alpha}{1+\alpha}\Sigma_1 \tag{10}$$

Note that $(\mu^*, \Sigma^*)$ are functions of the prior $(\mu_1, \Sigma_1)$, and so we used the function notation in (9–10) together with dependency on $S$ although $\mu^*$ does not depend on $\Sigma_1$ and $\Sigma^*$ similarly. We treat $(\mu^{SGD}, \Sigma^{SGD})$ as constant.

Next we define the meta prior ($\pi$) and posterior ($\rho$) distributions for the meta learning PAC-Bayes bound. As suggested in (Zakerinia et al., 2024),

$$\pi(A) = \mathcal{N}((\mu_1, \Sigma_1); 0, \kappa_\pi^2 I_{2d}) \tag{11}$$

$$\rho(A) = \mathcal{N}((\mu_1, \Sigma_1); M_1, \kappa_\rho^2 I_{2d}) \tag{12}$$

where $M_1 = (M_1^\mu, M_1^\Sigma)$ is the parameters for the meta posterior $\rho$. Here $\kappa_\pi$ and $\kappa_\rho$ are the fixed scales, and in their paper they used $\kappa_\pi = 100$ and $\kappa_\rho = 10^{-3}$. Finally, they assume delta meta prior and posterior in the complexity term: $\mathbb{P}(A) = \mathbb{Q}(A) = \delta_{P_1}$.

This leads to the following meta training objective:

$$\min_{M_1} \ \hat{R}(\rho)+$$

$$\underbrace{\sqrt{\frac{\text{KL}(\rho||\pi) + \log(4\sqrt{N}/\epsilon)}{2N}}}_{=:T_1} + \tag{13}$$

$$\underbrace{\sqrt{\frac{\text{KL}(\rho||\pi) + N\hat{E}(\rho) + \log(8nN/\epsilon) + 1}{2nN}}}_{=:T_2}$$

where

$$\hat{R}(\rho) = \mathbb{E}_{A\sim\rho}\left[\frac{1}{N}\sum_{i=1}^{N} l(A(S_i); S_i)\right]$$

$$\hat{E}(\rho) = \mathbb{E}_{A\sim\rho}\left[\frac{1}{N}\sum_{i=1}^{N} \text{KL}(A(S_i)||P_1)\right]$$

$$\text{KL}(\rho||\pi) = \frac{2d\kappa_\rho^2 + ||M_1||^2}{2\kappa_\pi^2} - d + 2d\log\frac{\kappa_\pi}{\kappa_\rho}$$

Recall that $n$ is the support data size and $N$ is the number of meta training tasks.

**Meta training procedure (stochastic approximation version).** Since it is nearly infeasible to compute the objective for all $N$ tasks (for the purpose of one gradient update for $M_1$), we adopt the stochastic approximate optimization. That is, for each sampled task batch $B$ (out of $N$ tasks), we compute the followings (stochastic estimates of the objective and the gradient):

- $\hat{T}_0 \approx \hat{R}(\rho)$ where

$$\hat{T}_0 = \frac{1}{|B|}\sum_{i\in B} l(\theta^{z_i}; S_i), \tag{14}$$

$$\theta^{z_i} = \mu^*(\mu_1, S_i) + \sqrt{\Sigma^*(\Sigma_1, S_i)} \cdot z_i, \ \ z_i \sim \mathcal{N}(0, I_d) \tag{15}$$

$$\mu_1 = M_1^\mu + \kappa_\rho \cdot z^\mu, \ \ z^\mu \sim \mathcal{N}(0, I_d) \tag{16}$$

$$\Sigma_1 = M_1^\Sigma + \kappa_\rho \cdot z^\Sigma, \ \ z^\Sigma \mathcal{N}(0, I_d) \tag{17}$$

  Here $\mu^*(\mu_1, S_i)$ and $\Sigma^*(\mu_1, S_i)$ are determined by (9–10). Note that $\hat{T}_0$ is a function of $M_1$, and the gradient of $\hat{T}_0$ with respect to $M_1$ can be obtained from $\nabla_{\theta^z}\hat{T}_0$ by the chain rule similarly as described previously.

- $T_1$ as it is since it is easy closed-form.

- $\hat{T}_2 \approx T_2$ where $\hat{T}_2$ only replaces $\hat{E}(\rho)$ by the following from $T_2$:

$$\hat{E}(\rho) \rightarrow \frac{1}{|B|}\sum_{i\in B} \text{KL}(A(S_i) \ || \ P_1) \tag{18}$$

$$= \frac{1}{|B|}\sum_{i\in B} \text{KL}(\mathcal{N}(\mu^*, \Sigma^*) \ || \ \mathcal{N}(\mu_1, \Sigma_1)) \tag{19}$$

  This has a closed form from the Gaussian KL formula.

As the above steps allow us to compute the meta training objective and its gradient, we can update $M_1$, and move on to a next batch.

**Test-time bound.** Once we have meta-learned $M_1$, we can compute the error bound for a new task that comes with the support data $S^*$. In essence, we solve the following optimization problem:

$$Q^* = \arg\min_Q \hat{R}(Q) + \sqrt{\frac{\mathrm{KL}(Q||P_1) + \log(8n/\epsilon) + 1}{2n}} \tag{20}$$

We parametrize $Q = \mathcal{N}(\mu, \Sigma)$ and solve it with respect to $(\mu, \Sigma)$. First, $\hat{R}(Q)$ can be approximated by $l(\theta^z; S^*)$ with $\theta^z = \mu + \sqrt{\Sigma} \cdot z$, $z \sim \mathcal{N}(0, I_d)$. We can use the chain rule similarly for $\nabla_{\mu, \Sigma} \hat{R}(Q)$. In the KL term, $P_1$ has to be sampled from $\rho$ (i.e., our learned $M_1$). We can use $P_1 = \mathcal{N}(M_1^\mu + \kappa_\rho \cdot z^\mu, M_1^\Sigma + \kappa_\rho \cdot z^\Sigma)$ and compute the KL using the Gaussian KL formula. Once optimized, the minimum value of (20) becomes the error bound for this test task which holds with probability at least $1 - \epsilon$.

# B  ARCHITECTURE AND TRAINING RECIPES

## B.1  DIFFUSION ARCHITECTURE AND TRAINING RECIPE

First, the diffusion model forwad encoder uses a 1000 timesteps with a linear scheduler over noise between 1e-4 to 2e-2. For the decoder, we use an MLP network for the diffusion model with 3 hidden-layers. The hidden layer dimension is $4\times$ the size of it's input. This is 10,240 for LaMP (divisible by 512 for parallelization concerns) and 4096 for vision. A layer conditioned time embedding of the diffusion step is added (summed with) the hidden layer's hidden representation.

The time embedding is generated from the diffusion timestep using sinsusoidal embeddings as per the original DDPM model (Ho et al., 2020). The dimensionality of the sinusoidal embedding is equal to the Diffusion MLP hidden dimension. Subsequently, the embedding is transformed using a two-layer MLP with first layer expanding the dimension to $4\times$ the network's hidden dimension and the second layer downscaling again to original hidden dimension. For example, on the vision experiments, the time embedding network has hidden dimension of $4098 \times 4$. Finally, to condition the time embedding computed by the two-layer MLP time embedding network for each layer, we apply a different linear transformation per diffusion hidden layer.

We train the diffusion model for 30K epochs for all experiments with a batch size of 1,024. We use the LAMB optimizer (You et al., 2020) with a learning rate of 0.01. For vision experiments, we found that we can continue improving performance if we continue training for a second stage of 10K more epochs. For the second stage, we use a one-cycle learning rate scheduler (Smith and Topin, 2018) with default hyperparameters (as found in pytorch). The maximum learning rate starts from 0.0004 reaching a maximum of 0.001 over a 1000 steps. We keep an exponential-moving average of the network weights throughout training with a decaying rate of 0.9999.

## B.2  FLAN-T5 + LAMP HYPERPARAMETERS

For training the base model across all datasets, we use LaMP's original recipe (Salemi et al., 2024). We use a batch size of 64, AdamW optimizer with a learning rate of 5e-5 and weight decay of size 0.0001. We use a linear warmup for the learning rate over 5% of the total number of training steps.

To build the model zoo, we found that we required to tune Adam optimizer learning rate and per-dataset epochs per-dataset. We optimizer the hyperparameters to improve performance on the training split (seen users) query data. We use learning rates of 0.01, 0.01, 0.0001 and 20, 10, 10 epochs for LaMP-2, LaMP-3 and LaMP-5 respectively. For all datasets, we used a linear warmup for the learning rate over 5% of the total number of training steps. We used the same recipe to train a per-user LoRA-XS model on the unseen users/novel tasks.

For LoRA-Hub, we used default hyperparameters as proposed by the original authors. First, we sample 20 random adapters from the model zoo. The weights of the linear combination is initialized with zeros and truncate min/max weights to -1.5/1.5. We do maximum inference steps of 40 with NeverGrad default hyperparameters.

Finally, we transform the SGD number of epochs to MeZO. The authors used $32\times$ as many epochs as SGD in the original paper (Malladi et al., 2023). This translates to 640, 320, 320 epochs for LaMP-2,

LaMP-3, and LaMP-5. We tested the three learning rates proposed to search over by the authors. We fixed a learning rate of 1e-3 across all datasets because we consistently found 1e-4 to not learn and 1e-2 to be unstable.

For LaMP, we sample 10K LoRA-XS modules from the diffusion model. We use k-means clustering on the diffusion samples to produce N clusters where N is chosen as the minimum Silhouette score. We evaluate N between [2,150] inclusive. For each cluster, we find the medoid; the adapter closest to the centroid of the cluster. During evaluation, we choose a cluster and evaluate all adapters therein. From the cluster, we short-list the best 15 adapters using the Flan-T5 training loss. On the best 15 adapters, we use text generation to produce an answer with greedy sampling. Using the LaMP benchmark proposed model selection metric for each dataset, we select the "winning" adapter.

### BOUND METRICS

For support errors, we use 1-Accuracy for LaMP-2, and ROUGE-1 for LaMP-5. For LaMP-3, both RMSE and MAE are not bounded. Therefore, we devise a cross-entropy like metric for the dataset. First, we convert the ordinal vectors to one-hot encodings. Subsequently, we calculate the absolute error between the labels one-hot encoding and Flan-T5 model logits, and divide by 2. This guarantees the error to be bounded in the [0-1] range inclusive. We use this metric as support error term.

**LaMP-2 Intricacies:** LaMP benchmark has one query sample per user. For LaMP-3 and LaMP-5, this suffices since the generated support error is continuous. Nevertheless, for LaMP-2, the accuracy term, which we use to derive the support error, becomes the 0/1 loss. Therefore, we split the support data in novel tasks to support and query data with ratio 80% and 20% respectively. If the split generates only 1 query samples, we move one sample from support to query to have a minimum of two-samples in query. We use the same split for all evaluations across SGD, MeZO, LoRA-Hub and our proposed methods. For reproducibility, all splits were done deterministically. Furthermore, we did not constrain the split to have same classes across both support and query. LaMP classification tasks are long-tailed. Therefore, for a novel task, a user might have classes X and Y in support but the query ends up with classes A and B making it a more challenging benchmark for all methods.

Finally, we truncate the support sizes of LaMP-3 and LaMP-5 to 256 samples across all methods. This is done deterministically for reproducibility. The reason for truncating the dataset is pure computational concerns.

### MODEL ZOO SIZE

For LaMP, we build a model zoo by training one PEFT adapter per-task in the dataset. Each user is treated as one task. This yields 3820, 20,000, and 9,682 total adapters/tasks in LaMP-2, LaMP-3, and LaMP-5 respectively.

### COMPUTE RESOURCES

All experiments were conducted on 10xA40 GPUs. The base model finetuning for generating task specific models took a couple of hours with a batch size of 64 using 2 GPUs. Time to train an adapter for one user using one GPUs varies from dataset to dataset and from user to user depending on their available dataset size. This takes as low as 10 seconds and up to 2 mins per user. Diffusion model training on the collected model zoo varied among datasets too since each epoch was of different total number of steps because of different dataset sizes. We used one A40 GPU per diffusion model and train using the recipe described earlier. Finally, downstream adaptation using STEEL varies between datasets. For LaMP-2, this was 50 A40 GPU hours and LaMP-5 was the slowest with 1200 A40 GPU hour since it requires text generation.

### B.3 CLIP + CoOP HYPERPARAMETERS

To build the model zoo, we used the authors original hyperparameters to train CoOP because we found them to work the best. This is SGD with a learning rate of 0.002. For Flowers-101, we train for 200 epochs. For DTD, FGVCAircraft and CUBirds, we trained for 300 epochs and found a One Cycle learning rate useful to stabilize training. These same hyperparameters were used to evaluate SGD on novel tasks. For BBPT, we use the authors default hyperparameters (Yu et al., 2023). We

found that the method converges within 8000 "API call". We attempted to run for a budget of 20K as our diffusion model offers but found that performance did not improve. Please note that the original authors reduce the dimensionality of the prompt using a small network because evolutionary optimization struggles in high-dimension. They reduce dimensionality to 512. Nevertheless, since we train only 2-tokens (dimensionality=1,024), then we do not use the small dimensionality reduction network.

Finally, for MetaPB, we train the upstream model for 20 epochs on our model zoo which takes 24 hours similarly to our diffusion model training. We use Adam with default learning rate of 0.01 for both upstream and downstream. Downstream adaptation was done for 300 epochs with same Adam optimizer setting. We train both means and variances. Variances are clipped to 0.1 for numerical stability.

For vision experiments, we found that exhaustive search was fast enough even though we sample 20K adapters from the diffusion model.

**Bound Metrics:** we use $1 -$ Accuracy as the support error term in our bound calculation for all vision experiments.

**Model Zoo Size:** We randomly sample 16-shot 5-way tasks for building the model zoo from each respective dataset. We train 10,000 total tasks per-dataset for the model zoo. The diffusion model is trained on this model zoo. Once trained, we sample the diffusion model once and fix the samples across all downstream evaluation for 1, 2, 4, 8 and 16 shots.

**Compute Resources:** For SGD, training one task upstream for model zoo collection or downstream for gradient descent adaptation takes 10-15 seconds. BBPT downstream takes 10-15 seconds as well. Training the diffusion model on a model zoo, since we fix model zoo size in CLIP experiments, was the same among all experiments. Each diffusion model took 24 A40 GPU hours to train. STEEL takes 30 second to 1 minute downstream for adapting to a new task. Finally, meta-training phase of MetaPB takes 24 hours similarly to our diffusion model training. MetaPB downstream takes 15 second per task too.

## C  EXTRA VISION RESULTS

### C.1  CERTIFYING ZERO-SHOT PERFORMANCE

Table 3: Comparing median bounds between STEELand zero-shot CLIP given a 16-shots support set.

| Median Bound | Flowers | CUB | DTD | FGVVCA |
|---|---|---|---|---|
| STEEL | 0.40 (+0.03) | 0.36 (-0.06) | 0.42 (+0.11) | 0.61 (+0.10) |
| ZS CLIP | 0.43 | 0.30 | 0.53 | 0.71 |

One might assume that zero-shot CLIP is a strong *certified* baseline because it is evaluated without any fine-tuning, and the support and query sets are drawn i.i.d. from the same distribution. This can lead to the impression that its support-set accuracy reflects test-time performance. However, these are still empirical estimates and do not account for uncertainty due to sampling. In contrast, our certificates explicitly bound the generalization gap from limited support data. To facilitate a fairer comparison, we compute a Langford-style test-set bound (Theorem 3.3 by Langford (2005)) using zero-shot CLIP's support accuracy from Table 3, treating it as a proxy for true performance. The resulting guarantees are included in Table C.1. Although zero-shot CLIP performs well empirically, our STEEL method achieves tighter guarantees and often stronger accuracy in the few-shot setting.

### C.2  ACCURACY RESULTS ACROSS SHOTS

Results are provided in table 4.

### C.3  VISION RESULTS WITH STANDARD ERRORS (STE)

We report the performance of the 16-shot 5-way vision experiments in Table 5. For the non-vacuous ratio, average gap, and median bounds, we compute the standard error using the bootstrap method.

Table 4: CLIP+CoOp few-shot learning. Accuracies over different number of shots.

| Dataset | Zero Shot | SGD | BBPT | Meta-PB | Model Zoo | STEEL |
|---|---|---|---|---|---|---|
| 1-Shots | | | | | | |
| CUBirds | 83.82% | 81.77% | 85.80% | 88.25% | 86.00% | 86.72% |
| DescribableTextures | 67.29% | 69.97% | 74.10% | 78.55% | 72.12% | 73.97% |
| FGVCAircraft | 47.44% | 53.00% | 55.05% | 56.46% | 54.37% | 55.40% |
| Flowers-101 | 81.14% | 83.55% | 80.40% | 54.20% | 74.25% | 76.75% |
| Avg | 69.92% | 72.07% | 73.84% | 69.37% | 71.69% | 73.21% |
| 2-Shots | | | | | | |
| CUBirds | 83.82 | 85.82% | 86.92% | 88.40% | 86.77% | 86.50% |
| DescribableTextures | 67.29 | 75.95% | 76.57% | 78.99% | 75.22% | 76.17% |
| FGVCAircrafts | 47.44 | 52.30% | 57.90% | 58.31% | 55.82% | 57.30% |
| Flowers-101 | 81.14 | 86.92% | 85.05% | 61.25% | 77.17% | 79.50% |
| Avg | 69.92 | 75.25% | 76.61% | 71.74% | 73.75% | 74.87% |
| 4-Shots | | | | | | |
| CUBirds | 83.82 | 88.30% | 88.42% | 88.77% | 87.50% | 87.47% |
| DescribableTextures | 67.29 | 81.02% | 77.90% | 78.85% | 77.40% | 79.82% |
| FGVCAircrafts | 47.44 | 58.65% | 60.55% | 58.26% | 57.55% | 60.10% |
| Flowers-101 | 81.14 | 91.95% | 87.07% | 65.44% | 79.82% | 81.42% |
| Avg | 69.92% | 79.98% | 78.49% | 72.83% | 75.57% | 77.21% |
| 8-Shots | | | | | | |
| CUBirds | 83.82 | 89.75% | 88.40% | 88.68% | 87.42% | 87.32% |
| DescribableTextures | 67.29 | 85.07% | 81.47% | 79.96% | 79.90% | 81.77% |
| FGVCAircrafts | 47.44 | 62.07% | 61.55% | 58.56% | 58.77% | 61.72% |
| Flowers-101 | 81.14 | 94.30% | 88.67% | 68.72% | 80.32% | 82.52% |
| Avg | 69.92% | 82.80% | 80.02% | 73.98% | 76.61% | 78.34% |
| 16-Shots | | | | | | |
| CUBirds | 83.82 | 90.32% | 89.27% | 89.24% | 87.97% | 88.40% |
| DescribableTextures | 67.29 | 87.95% | 83.20% | 81.12% | 79.25% | 81.50% |
| FGVCAircrafts | 47.44 | 65.57% | 62.37% | 59.84% | 61.02% | 61.37% |
| Flowers-101 | 81.14 | 95.90% | 90.15% | 71.23% | 82.92% | 84.90% |
| Avg | 69.92% | 84.94% | 81.25% | 75.36% | 77.79% | 79.04% |

Specifically, we sample with replacement and report the standard deviation of the resulting distribution as the estimated standard error. For accuracy, computing STE is straightforward.

## D    STEEL PSEUDO-CODE

We provide pseudo-code in Algorithm D. There are upstream and downstream learning phases, as per meta-learning, model-zoo, and other transfer learning workflows. We refer to these as meta-training and meta-testing, to borrow terminology from the meta-learning literature Hospedales et al. (2021); Wang et al. (2020). The upstream meta-training phase trains one model per-task and then treats these models as data to learn a weight-space diffusion model. This effectively compresses upstream models into a single model generator that provides a prior finite hypothesis class. The downstream meta-testing phase performs low-shot learning by sampling from the generator and returning the model that minimises the downstream empirical risk. This leads strong generalisation guarantees in practice, because the finite hypothesis class incurs a low complexity cost in the certificate calculation.

[H] Transfer Learning via Diffusion and Finite Hypothesis Space

**Meta-Training Phase:** [1] **Input:** Training tasks $\{T_1, T_2, \ldots, T_N\}$ sampled i.i.d. from $p_{\text{true}}(T)$
**Output:** Trained diffusion model $p(\theta)$

Table 5: Aggregate over 16-Shots 5-way visual recognition episodes with Standard Errors

| Method | SGD | BBPT | Meta-PB | STEEL |
|---|---|---|---|---|
| **CUBirds** | | | | |
| Non-Vacuous Ratio | 0.00 ± 0.00 | 0.00 ± 0.00 | 97.50 ± 2.47 | 100.00 ± 0.00 |
| Average Gap | 2.49 ± 0.01 | 2.48 ± 0.01 | 0.45 ± 0.01 | 0.24 ± 0.01 |
| Min Bound | 2.55 | 2.55 | 0.34 | 0.30 |
| Median Bound | 2.58 ± 0.01 | 2.59 ± 0.01 | 0.55 ± 0.01 | 0.36 ± 0.01 |
| Max Bound | 2.64 | 2.68 | 0.85 | 0.47 |
| Average Accuracy | 90.32 ± 0.78 | 89.27 ± 0.50 | 89.24 ± 0.70 | 88.40 ± 0.72 |
| **Describable Textures** | | | | |
| Non-Vacuous Ratio | 0.00 ± 0.00 | 0.00 ± 0.00 | 50.00 ± 7.91 | 100.00 ± 0.00 |
| Average Gap | 2.43 ± 0.01 | 2.46 ± 0.01 | 0.54 ± 0.01 | 0.24 ± 0.01 |
| Min Bound | 2.55 | 2.55 | 0.57 | 0.36 |
| Median Bound | 2.55 ± 0.00 | 2.63 ± 0.01 | 0.80 ± 0.02 | 0.42 ± 0.01 |
| Max Bound | 2.58 | 2.78 | 1.06 | 0.53 |
| Average Accuracy | 87.95 ± 0.58 | 83.20 ± 0.76 | 81.12 ± 0.68 | 81.50 ± 0.96 |
| **FGVCAircrafts** | | | | |
| Non-Vacuous Ratio | 0.00 ± 0.00 | 0.00 ± 0.00 | 0.00 ± 0.00 | 97.50 ± 2.47 |
| Average Gap | 2.31 ± 0.01 | 2.45 ± 0.01 | 0.86 ± 0.02 | 0.22 ± 0.01 |
| Min Bound | 2.58 | 2.64 | 0.87 | 0.45 |
| Median Bound | 2.64 ± 0.01 | 2.80 ± 0.02 | 1.25 ± 0.03 | 0.61 ± 0.02 |
| Max Bound | 2.78 | 3.01 | 1.76 | 0.85 |
| Average Accuracy | 65.57 ± 1.45 | 62.37 ± 1.60 | 59.84 ± 1.21 | 61.37 ± 1.25 |
| **Flowers-101** | | | | |
| Non-Vacuous Ratio | 0.00 ± 0.00 | 0.00 ± 0.00 | 10.00 ± 4.74 | 100.00 ± 0.00 |
| Average Gap | 2.51 ± 0.00 | 2.50 ± 0.01 | 0.68 ± 0.03 | 0.27 ± 0.01 |
| Min Bound | 2.55 | 2.55 | 0.65 | 0.31 |
| Median Bound | 2.55 ± 0.00 | 2.59 ± 0.01 | 1.15 ± 0.10 | 0.40 ± 0.02 |
| Max Bound | 2.55 | 2.70 | 1.77 | 0.61 |
| Average Accuracy | 95.90 ± 0.46 | 90.15 ± 0.95 | 71.23 ± 2.40 | 84.90 ± 1.43 |

$i = 1$ to $N$ Train PEFT adapter on task $T_i$ to obtain optimal parameters $\theta_i$ $\overline{\Theta} \leftarrow \overline{\Theta} \cup \{\theta_i\}$ Collect task-specific parameters

Train diffusion model $p(\theta)$ using parameter set $\overline{\Theta} = \{\theta_1, \theta_2, \ldots, \theta_N\}$ as training data **Meta-Testing Phase:** [1] **Input:** New task $T^* \sim p_{\text{true}}(T)$ with support set $S^* = \{(x_i, y_i)\}_{i=1}^{n}$ **Input:** Trained diffusion model $p(\theta)$, hypothesis space size $|\Theta|$ **Output:** Selected parameters $\theta^*$ and generalization bound

Sample finite hypothesis space: $\Theta = \{\theta^{(1)}, \theta^{(2)}, \ldots, \theta^{(|\Theta|)}\} \sim p(\theta)$

**Evaluate-then-Select Strategy:**
Can use hierarchical search $j = 1$ to $|\Theta|$ Compute empirical risk: $r(\theta^{(j)}) = \frac{1}{n} \sum_{(x,y) \in S^*} l(\theta^{(j)}; x, y)$

Select optimal parameters: $\theta^* = \arg\min_{\theta \in \Theta} r(\theta)$

**Compute Risk Certificate:** With probability $\geq 1 - \epsilon$, the generalization error satisfies: $R(\theta^*) \leq r(\theta^*) + C \cdot \sqrt{\frac{\log \frac{|\Theta|}{\epsilon}}{2n}}$

# E    DIFFUSION MODEL ABLATION

The use of a diffusion model in STEEL is motivated by recent work on generative modeling in weight space (Wang et al., 2024), which demonstrates that diffusion models can effectively capture the complex, multimodal distribution of neural network parameters. To evaluate whether this modeling capacity is critical for performance, we conduct an ablation study comparing against two alternative generative baselines for sampling parameter candidates in the LoRA-XS subspace:

- **Random Sampling:** Each parameter is sampled independently from a Gaussian distribution whose mean and standard deviation are estimated from the corresponding statistics of the model zoo. This baseline assumes a fully factorized, unimodal distribution and requires no training or hyperparameter tuning.
- **Gaussian Mixture Model (GMM):** A GMM with diagonal covariance matrices is trained on the model zoo parameters. The number of mixture components is selected from $\{2, 4, 8, 16, 32, 64\}$ using the Bayesian Information Criterion (BIC). Once trained, the GMM is used to sample candidate parameter vectors.

Table 6: Comparison of generative models for sampling parameter candidates on LaMP-2. The diffusion model is the only method that improves upon both the zero-shot baseline and SGD fine-tuning.

| Method | Zero-Shot | SGD | Random | GMM | Diffusion |
|---|---|---|---|---|---|
| Accuracy | 61.68% | 63.25% | 0.00% | 60.29% | **63.74%** |

The results in Table 6 demonstrate that the diffusion model significantly outperforms both random sampling and GMM-based sampling. Random sampling performs poorly, with zero-accuracy. Generated data from this baseline are random tokens. GMM improves over random but still fails to outperform the zero-shot baseline, likely due to its limited capacity to model high-dimensional parameter correlations and multimodality. In contrast, the diffusion model exceeds the performance of SGD-based fine-tuning, despite requiring no training on the test-time user data. This highlights its strong generative prior over parameter space.

## F  BROADER IMPACT

This paper presents work whose goal is to advance the field of Machine Learning. There are many potential societal consequences of our work, and we mention a non-exhaustive concerns here. First, for LLM experiments, please note that the diffusion model is trained on adapters trained on user-specific data. We can use the diffusion model to interpolate between adapters to potentially avoid distributing the private model zoo among clients, but this does have potential for data leakage by memorization Staab et al. (2024). It is worth investigating privacy preserving methods for such concerns Miranda et al. (2024).

On the vision side of our experiments, despite our benchmark being on fine-grained classification tasks of publicly available datasets, we necessitate the reminder of ethical concerns in computer vision as our method is easily applicable across domains and applications Waelen (2023).

