# OpenReview forum: "Model Diffusion for Certifiable Few-shot Transfer Learning"
_ICLR.cc/2026/Conference — Submitted to ICLR 2026_

### Official Review · Reviewer_ztW4 · 2025-10-23

**Soundness:** 2
**Presentation:** 2
**Contribution:** 2
**Rating:** 4
**Confidence:** 3

**Summary:**

This study try to use diffusion model to learn the distribution of various tasks under the brief that the any task can be identified by the model's parameters. Based on the learned diffusion model, the authors claimed that this distribution can reduced the complexity of hypothesis space from infinite to finite, thereby resulting in a tighter generalization bound, especially under the scenario of a few samples. The authors subsequently conducted abundant experiments to verified their idea.

**Strengths:**

1. The idea of using a diffusion model to learn the distribution of tasks and then applying it to transfer learning sounds like a interesting idea. Even though it requires a strong assumption that the task can be totally identified by model parameters with little or no ambiguity.
2. The performance of STEEL is powerful across various few-shot datasets compared to the other methods.

**Weaknesses:**

I have several concerns about STEEL
1. The authors use a diffusion model to learn the potential distribution of "tasks" and then sample a collection of candidate $\theta_i$ from it. However, this sampling process does not appear to leverage any information specific to the target task. As a result, STEEL essentially employs a diffusion-based method to reduce the hypothesis space, rather than directly imposing a meaningful prior constraint informed by the target task itself. While reducing the size of the hypothesis space can indeed tighten the generalization bound, it may also diminish the expressive power of the model. There is no evidence provided that the optimal parameter for the target task remains within the reduced hypothesis space, especially in the absence of any target-specific information. From this perspective, I am concerned that the motivation of this paper may be stereotype. In other words, if the authors focus solely on achieving a tighter generalization bound without considering the impact of hypothesis space richness, it becomes trivial to obtain a minimal upper bound simply by excessively restricting the hypothesis space—for example, to a linear or even a constant model. However, such an approach would also severely compromise the expressive capacity of the model.

2. In the initial version, the author didn't compare the cost during both training process and testing process with other methods. I suppose it will enhance the solidness of this paper but doesn't influence the core impact of this study.

**Questions:**

Pls see Weaknesses

**Summary of Review**: As elaborated in the first item of Weaknesses, I do not think the authors' motivation is sufficiently reasonable, as the approximation ability of the hypothesis space is overlooked. While this omission is sometimes accepted in learning theory, where the focus is primarily on statistical aspects, it is not appropriate when discussing the size of the hypothesis space, given the strong connection between model performance and the richness of the hypothesis space. Nevertheless, considering that this study is not in the theory track and the authors provide improved practical results, I am inclined to give a score of 4.

---

> ### Author Response · Authors · 2025-11-21
>
> - W1: Sampling process does not leverage information specific to the target task. STEEL reduces the hypothesis space rather than imposing a prior informed by the target task.
>
> The reviewer is correct that we do not impose a prior informed by the target task, but mistaken in that  this is not a problem. Basic practice in learning theory is to define the hypothesis class or prior before observing the target task. Making the hypothesis class training-data dependent complicates generalization guarantees and risks overfitting.
>
> Rather than trying to make the hypothesis class target class dependent, our strategy to provide a good hypothesis class takes inspiration from model zoos and meta-learning (surveys: Wang 2020, Hospedales 2021). Specifically, we follow this literature in studying problems where there is a family of related tasks (e.g., Many users for which an LLM should be personalised in our LaMP experiments) which we can divide into upstream tasks for hypothesis class learning, and downstream tasks that can be better solved using the hypothesis class customized upstream for this family of problems.
>
> Thus STEEL reduces the hypothesis space by finding a compact hypothesis class that can solve the (seen) upstream tasks well, and then re-using it to solve the (unseen) downstream tasks. In this sense it is exactly the same problem setting and high-level strategy of competitors such as MetaPB (Zakerinia ICML’24). MetaPB transfers a compact but continuous prior distribution from the upstream to downstream tasks. STEEL’s innovation is to transfer a finite hypothesis class from upstream to downstream tasks.
>
> - W1: There is no evidence provided that the optimal parameter for the target task remains within the reduced hypothesis space.
>
> Evidence for this is provided throughout the paper.
>
> The most explicit evidence appears in Fig 4 and Fig 5, which show curves for support error. Support error quantifies the empirical risk on the downstream training set—that is, it measures the extent to which the transferred finite hypothesis class can explain the training data of the downstream tasks. The low support error demonstrates that good solutions to the downstream tasks do indeed lie within our reduced finite hypothesis space.
>
> Less explicitly, but even more decisively, all our results measure certificate quality, defined as empirical risk (support error) + complexity (Eq 3). Good empirical risk is a necessary condition for a good certificate. If no good solution to the target task existed within our hypothesis space, support error would be high and the certificates would be vacuous. The fact that we achieve strong certificates throughout (Fig 2, Tab 1, Tab 2, etc.) is direct evidence that good solutions to the target tasks lie within our novel hypothesis class.
>
> Finally, it's worth noting that in the low-shot regime, the "optimal" target task solution in the ERM sense would achieve 100% training accuracy but likely 0% generalization, as it would completely overfit the downstream data. Our goal is not to "optimally" fit the target task, but to fit it well enough for strong generalization—which we demonstrate and provably certify throughout.
>
> - W1: If the authors focus solely on achieving a tighter generalization bound without considering the impact of hypothesis space richness, it becomes trivial to obtain a minimal upper bound simply by excessively restricting the hypothesis space—for example, to a linear or even a constant model. However, such an approach would also severely compromise the expressive capacity of the model.
>
> We agree that excessively restricting the hypothesis space can trivially prevent overfitting at the cost of increased empirical risk (failed support set fitting). However, this is not happening in our case.
>
> First, the hypothesis space is not excessively restricted. We obtain good support error (Fig 4 & 5), showing that the transferred finite hypothesis class adequately explains the downstream training data. Second, "compromised expressive capacity" (poor empirical risk) would manifest as vacuous bounds in our certificates, since certificate quality depends on the sum of empirical risk and hypothesis class complexity (Eq 3). Our consistently strong certificates (Tab 1 & 2) demonstrate good empirical risk throughout.
>
> STEEL avoids these pitfalls with the same high-level workflow as related methods like MetaPB and LoRAHUB: provide a constrained hypothesis class well-suited to the task family (determined on upstream tasks) that transfers to fit downstream tasks. The hypothesis class must be expressive enough to fit downstream training sets while constrained enough to generalize well. The key difference is that STEEL's novel discrete hypothesis class achieves a far better expressiveness-complexity tradeoff than prior methods, yielding substantially better certificates.
>
> - W2.
>
> Please see the response to the reviewer GiGA. Our method is faster than MeZo, and comparable to MetaSGD and MetaPB.

---

### Official Review · Reviewer_GiGA · 2025-10-26

**Soundness:** 2
**Presentation:** 2
**Contribution:** 2
**Rating:** 4
**Confidence:** 3

**Summary:**

The paper studies low-shot cross-task transfer learning. In the upstream phase, the method fits trainable modules to available source tasks and then trains a parameter diffusion model to generate models according to the task distribution. In the downstream phase, it follows a sample-then-evaluate paradigm. However, the architecture of the diffusion model and its training procedure are not clearly described. Experimental results show improved bounds and non-trivial generalization guarantees.

**Strengths:**

The paper provides some theoretical analysis, and the experiments indicate improved bounds and non-trivial generalization guarantees.

**Weaknesses:**

1. The title in the submitted PDF differs from the one on OpenReview, which raises concerns.
2. The main contribution appears to be learning a parameter diffusion model to generate PEFTs according to the task distribution. However, the architecture and training of the diffusion model are unclear. Section 3.3 discusses its role only at a high level, stating it is trained on PEFT parameters {θi}, but omits key architectural details and training objectives. The paper seems to assume readers are already familiar with parameter diffusion models, which may not be the case. While I am familiar with generalization theory and image/language diffusion models, I could not find sufficient background or specific descriptions of the parameter diffusion model used here.
3. In terms of task performance, the method does not consistently or significantly outperform baselines across Tables 1 and 2, even though it achieves better bounds and non-trivial generalization guarantees.
4. Efficiency concerns: The method requires evaluating thousands (sometimes tens of thousands) of candidate hypotheses per support set, which is computationally expensive and may be impractical outside research settings. Although Section 3.4 mentions hierarchical search and clustering, there is no detailed comparison between exhaustive and approximate search strategies.

**Questions:**

See weaknesses above.

---

> ### Author Response · Authors · 2025-11-21
>
> We thank the reviewer for the detailed feedback. We address each concern below and hope to clarify some misunderstandings about the paper's goals and experimental design.
>
> 1. **Difference in titles**
> - We revised the paper title during preparation of the PDF but inadvertently forgot to update the metadata on OpenReview, leading to the mismatch. This was an oversight. Conceptually, the diffusion-based approach we use is a standard instance of weight-space learning, so the two titles refer to the same underlying methodological framing. We will ensure the PDF and OpenReview titles are fully consistent in the camera-ready version.
>
> 2. **Implementation details**
> - We thank the reviewer for pointing out that Section 3.3 did not provide sufficient architectural and training details for our parameter-space diffusion model. These details are indeed included in **Appendix B (Section B.1: “Diffusion Architecture and Training Recipe”)**, where we describe the full architecture (MLP depth, hidden sizes, timestep-conditioning mechanism, sinusoidal embeddings, and per-layer conditioning), the training setup (noise schedule, optimizer, learning-rate schedules, EMA, batch sizes), and all dataset-specific configurations. We will make this clearer in the camera-ready version by adding explicit forward references in the main text
> - The reason for omitting these details is to keep the reader unfamiliar with risk certificates engaged at a conceptual and methodological level while deferring implementation details to the Appendices and code releases. We will release the codebase as well as model zoos, diffusion models, diffusion model samples used for inference, and experimental logs for all our experiments.
>
> 3. **STEEL empirical error**
> - Our goal here is to achieve better guarantees while retaining comparable empirical accuracy to baselines. (This is usually the goal of methods designed to produce good certificates, since there is usually some tradeoff between the two metrics, and it’s non-trivial to simultaneously strengthen the certificate and empirical accuracy). The results show this goal is achieved, as empirical accuracy is comparable to baselines, but our guarantees are dramatically better than baselines. In high-stakes applications, provable guarantees on performance are more valuable than a few % difference in empirical test accuracy.
>
> 4. **Efficiency concerns**
> - To clarity first our LaMP (language model) experiments use hierarchical search + clustering; while our vision tasks use exhaustive search over the model zoo.
> - Please note that our “thousands of candidate hypothesis evaluations” is not unreasonable. Recall that even in the simplest SGD baseline, the number of evaluations is proportional to the number of gradient-descent iterations, which could also be a large number. In practice the runtime overhead is comparable enough with the baselines. For example:
>   - **Vision downstream time per task:**
> | SGD | MetaPB | STEEL |
> |-----|--------|-------|
> | 10-15 sec | 15 sec | 30-60 sec |
>
>   - **Language downstream time per task:**
>
>     |        | SGD | STEEL |
>     |--------|-----|-------|
>     | LaMP-2 | 10s - 2 min | 0.8 min |
>
> - Some other competitors such as MeZo (Malladi 2023) are about 2.5x slower in wall-clock time than SGD, due to requiring many more epochs than SGD. Thus STEEL is clearly faster than MeZo, whose structure doesn’t allow for acceleration via hierarchical search.
> - With optimized infrastructure such as vLLM serving, which we did not use in our experiments so far, LaMP-5 adaptation drops  to seconds per user, making large-scale personalization highly practical for deployment.

---

### Official Review · Reviewer_dH3X · 2025-11-01

**Soundness:** 3
**Presentation:** 4
**Contribution:** 4
**Rating:** 8
**Confidence:** 3

**Summary:**

The paper proposes STEEL: learn a diffusion model over weight-space PEFT parameters $p(\theta)$ from a model-zoo of upstream adapters, and at test time sample a finite set $\Theta$ and evaluate–then–select the adapter with the lowest support loss. Casting few-shot adaptation as selection over a finite hypothesis set allows PAC/PAC-Bayes–style non-vacuous risk certificates, while keeping test accuracy broadly comparable to standard learners.

**Strengths:**

- A neat combination of weight-space generative modeling (DDPM) to form a finite hypothesis set and evaluate–then–select to keep the complexity term fixed while minimizing empirical risk—yielding non-vacuous certificates in few-shot regimes.
- Experimental breadth & rigor. Evaluations span multiple LaMP tasks and vision datasets under standard meta-learning protocols, reporting not only accuracy but also % non-vacuous, bound statistics, and gap.
- The paper visualizes how certification varies with $|\Theta|$ and demonstrates non-vacuous certificates from low-shot settings onward.
- The training of $p(\theta)$, sampling of $\Theta$, and the hierarchical search details are described clearly enough to reproduce.

**Weaknesses:**

(W1) The paper fixes LoRA-XS (~2.6K params) and CoOp (1,024-dim prompt) without varying LoRA rank or token count. Given both diffusion learnability and certification can depend on $\dim(\theta)$, a $\theta$-size sweep would strengthen the claims.

(W2) Fairness to model-zoo under matched search. Hierarchical search is an inference strategy, not unique to STEEL. A controlled comparison where model-zoo also uses the same k-means/medoid + top-15 pipeline would isolate the benefit of diffusion sampling vs search.

(W3) Choice of $|\Theta|$ vs. zoo size not probed. In vision, STEEL uses 20K samples while the zoo size is 10K. Results for $|\Theta|=10K$ (matched to the size of model-zoo) are not reported, leaving unclear whether gains come from diffusion coverage or simply larger candidate pools.

(W4) Missing reporting of the chosen cluster count $N$. For LaMP, $N$ is selected via minimum silhouette over a range, but the actual values (or distribution) and typical cluster sizes re-evaluated are not reported, making it hard to assess how much improvement stems from search granularity.

**Questions:**

Please provide responses to the reviews mentioned in the Weaknesses section.

---

> ### Author Response · Authors · 2025-11-21
>
> Thanks for the careful read and the accept! These are great questions about experimental fairness. We realize we didn't make it clear enough in the paper, but we actually already controlled for all of these factors in our experiments. We've added the specific details and numbers below.
>
> **(W1)**
> - As weight space learning methods improve and scale, the dimensionality of task adapters we can learn a distribution over will improve.
> - Please note that certification does not depend on the dimensionality of the adapter size but rather the count of the sampled adapters, and the latter is analyzed in Fig 4.
>
> **(W2)** We agree that this would be a strong confounder in experimental comparisons. Please be reassured that the reported performance on LaMP experiments employs hierarchical search for both model zoo and STEEL.
>
> **(w3)** Figure 4 varies the size of |theta|, and it shows that the difference in support loss between 10 and 20k is not substantial. To be more concrete, we show below the different in results between evaluating 10k and 20k adapters.STEEL outperforms model zoo whether or not we use 10k or 20k samples.
> | Dataset                | 10k zoo | 10k steel | 20k steel |
> |------------------------|---------|-----------|-----------|
> | CUBirds                | 87.97%  | 88.30%    | 88.40%    |
> | DescribableTextures    | 79.25%  | 81.42%    | 81.50%    |
> | FGVCAircrafts          | 61.02%  | 61.50%    | 61.37%    |
> | Flowers-101            | 82.92%  | 84.22%    | 84.90%    |
> | Avg                    | 77.79%  | 78.86%    | 79.04%    |
>
> **(W4)**
> - We will release model zoos, diffusion models, diffusion model samples used for inference, and logs for all our experiments alongside code. These logs includes our clustering analysis. To answer your question directly:
>   - Diffusion model: 22, 92 and 14 clusters for LaMP-2, 3, and 5 respectively
>   - Model zoo: 5, 150, and 54 clusters for LaMP-2, 3 and 5 respectively

---

### Official Review · Reviewer_c9SQ · 2025-11-09

**Soundness:** 3
**Presentation:** 3
**Contribution:** 3
**Rating:** 6
**Confidence:** 3

**Summary:**

This paper introduces a STEEL - a diffusion hypernetwork to produce PEFT weight candidates at test time in order to provide stronger / tighter generalization bounds. By sampling weights from the diffusion model, and selecting via sample-and-evaluate, risk certificates can be made tighter. On both LLMs and several few-shot vision tasks, the authors offer supporting practical evidence on a smaller scale, mostly matching normal PEFT performance while providing much better generalization bounds,

**Strengths:**

The paper is well written and straightforwward to follow; while STEEL itself is sensible - you utilize a diffusion model as a weight candidate generator for your test-time PEFT in order to provide tighter (and actual) generalization bounds. The combination of PEFT hypernetwork and generalization bounds is, to the best of my knowledge, novel - and sensible, with convincing results in 5.1 and 5.2 on both LLM and Vision-model adaptation.

**Weaknesses:**

I do think this paper provides a very interesting use of PEFT hypernetworks for generalization bounds, which to me comes with one major question / issue:

The whole approach hinges on L. 193: "We expect that Θ is rich enough to represent the true task distribution ptrue(T) faithfully, and the
adapted (“selected”) θ will generalize well on unseen samples from T", which is a very, very strong statement to make. By default, STEEL is likely much more limited when it comes to adapting to larger distribution shifts, as no weights are actually learned on support data (i.e. downstream data). This means that generalization bounds really only hold when the downstream data is much more i.i.d. with respect to the upstream data compared to normal PEFT, no? I.e. what does the performance look like as the downstream data moves progressively further away from the upstream data? At what point would be we a significant drop in performance of STEEL?
Or, from the opposite perspective - how well do these generalization bounds and comparable performances hold as the underlying mode is scaled up alongside the upstream data and the diffusion candidate generator?


Nitpick: Submission title does not match paper title

**Questions:**

See Weaknesses.

---

> ### Author Response · Authors · 2025-11-21
>
> We thank the reviewer for engaging with the core theoretical question of our approach and appreciate the chance to clarify the validity and strength of our assumptions.
>
> **The reviewer touches on two assumptions: (i) upstream-downstream IID, (ii) discrete hypothesis class (L193).**
>
> (i) The upstream-downstream IID assumption, or some analogy thereof, is **standard in meta-learning** and shared by all few-shot learning (FSL) methods including our baselines (MetaPB, LoRA-Hub, Model Zoo). All meta-learning frameworks (surveys in Wang 2020, Hospedales 2021) tackle FSL by assuming upstream and downstream tasks are drawn from the same task distribution, and some knowledge learned from the upstream tasks can generalise to downstream tasks.
>
> For example, MetaPB assumes a common prior exists across all tasks, and that this prior can be learned from upstream tasks and applied to downstream tasks. The seminal MAML (Finn ICML’17) assumes a common initial condition from which both upstream and downstream tasks can be solved by few-step fine-tuning. LoRA-Hub assumes a common dictionary of base models, from which any downstream task can be solved by linear interpolation.
>
> Learners in this family are therefore all mainly making claims about IID training + testing tasks. Similar to other learners in this family, our benchmarks are designed as such:
> - LaMP: Language tasks that are user-wise IID.
> - Vision benchmarks: Object recognition tasks with category-wise IID (such as recognising different species of flowers).
>
> (ii) The discrete hypothesis class assumption is fairly unique to our method. The reviewer suggests that it might be too strong to fit the downstream data well. However, this is empirically not the case in all our experiments. Please remember that our certificates are of the form Empirical Risk + Complexity (Eq 1, Eq 3). An overly restrictive hypothesis class that can’t fit the downstream data would show up as high empirical risk, and thus vacuous certificates. However, our results show reasonable empirical risk (see Support Error curve in Fig 4), and strong certificates that are better than state of the art competitors (Tab 1,2). So it’s not an overly restrictive hypothesis class. The fact that this surprisingly simple hypothesis class does a good job in practice, is a key novel insight of our paper!
>
> Additional Remark: All few-shot learners succeed by constraining the hypothesis class in some fairly strong way that prevents overfitting. EG: MAML only considers models in a local region around the upstream learned initial condition. LoRA-Hub constrains to a low-dimensional linear subspace spanned by the task library. STEEL is more constrained than these in the sense that it considers a finite hypothesis set. But it is actually less constrained in that it can represent a multi-modal task distribution, and need not be a locally contiguous set of hypotheses like MAML and LoRA-Hub. In this sense STELL could actually represent a more diverse set of models than prior art.
>
> **TLDR** The IID assumption is reasonable and shared by all related methods. The discrete hypothesis assumption is unique, but results show it’s not overly strong.
>
> **W2 What happens if the task-wise IID assumption is violated?**
>
> One can challenge any meta-learning or model-zoo type algorithm on OOD task robustness, since they all make such an assumption. Empirically, we evaluated an OOD case between CUB and iNaturalist in Fig 5. The result shows that support error is still low, meaning the learned discrete hypothesis set was able to fit this mildly OOD set of downstream tasks; and the overall certificate is still good, even for OOD tasks.
>
> We emphasise that while our motivating intuition uses a task-level IID assumption to suggest why the empirical risk might be small, the PAC Bayes theorem we use (Eq 3) does not require IID tasks. Thus our framework will always produce correct certificates, whether or not the downstream tasks are IID or OOD. Even in an extreme case where the downstream tasks are so extremely OOD that the finite hypotheses cannot fit them, it will show up as a weak certificate that notifies the user that the results should not be trusted.
>
> With reference to the specific reviewer question *“This means that generalization bounds really only hold when the downstream data is much more i.i.d. with respect to the upstream data compared to normal PEFT?“* The answer is: (1) No. STEEL (thanks to PAC Bayes) will always produce a generalization bound that holds. In the case of severe OOD it will show up as a looser but still correct bound. (2) No. Normal PEFT has vacuous generalization bounds altogether in the low-shot regime, even for IID case. SotA generalization bounds for PEFT (eg Lotfi 2024) still require 10,000s of examples to obtain non-vacuous bounds even in the easy IID case. STEEL already has non-vacuous bounds with only 100s of examples, and this is demonstrated forfor at least one at least one moderately OOD case (Fig 5).

---

### Meta-Review · Area_Chair_MQRZ · 2026-01-08

**Summary:**

This paper proposes a diffusion‑based hypernetwork that samples parameter efficient fine tuning weight candidates at test time. They provide PAC-Bayes type generalization certificates in few-shot scenarios. I find the idea interesting and there is a nice empirical component on several vision datasets, but the paper does not meet the bar due to several remaining concerns
However, after considering all reviews and the author responses, the paper ultimately falls short of acceptance for several reasons. One gap in the motivation is that the hypothesis class is restricted without conditioning on downstream data, so it is unclear why and when such a compressed/reduced class is expressive enough under distribution shift. Some details, most notably those related to the parameter‑space diffusion model which is a key component, are missing, and similarly there are some important gaps missing in the experimental section.

**Reviewer Concerns:**

The concerns outlined above are not clearly addressed to satisfaction during the rebuttal.

**Reviewer Scores:**

Reviewer scores are consistent with their reports and it is unclear the last two reviewers are sufficiently convinced after rebuttal

---

### Decision · Program_Chairs · 2026-01-26

Reject